# Water Body Identification from Satellite Images Using a Hybrid Evolutionary Algorithm-Optimized U-Net Framework

**DOI:** 10.3390/biomimetics10110732

**Published:** 2025-11-01

**Authors:** Yue Yuan, Peiyang Wei, Zhixiang Qi, Xun Deng, Ji Zhang, Jianhong Gan, Tinghui Chen, Zhibin Li

**Affiliations:** 1School of Computer Science and Technology, Chongqing University of Posts and Telecommunications, Chongqing 400065, China; yuanyue@cigit.ac.cn; 2Chongqing Institute of Green and Intelligent Technology, Chinese Academy of Sciences, Chongqing 400714, China; 3School of Software Engineering, Chengdu University of Information Technology, Chengdu 610225, China; zhixiangqi0905@163.com (Z.Q.); gjh@cuit.edu.cn (J.G.); chenth199208@163.com (T.C.); lizhibin111@outlook.com (Z.L.); 4Dazhou Key Laboratory of Government Data Security, Sichuan University of Arts and Science, Dazhou 635000, China; 5Chongqing Institute of Meteorological Sciences, Chongqing 401147, China; zhangji0324@163.com; 6Xinjiang Technical Institute of Physics & Chemistry, Chinese Academy of Sciences, Urumqi 830011, China

**Keywords:** water body identification, evolutionary algorithms, hyperparameter optimization, deep learning, remote sensing, semantic segmentation

## Abstract

Accurate and automated identification of water bodies from satellite imagery is critical for environmental monitoring, water resource management, and disaster response. Current deep learning approaches, however, suffer from a strong dependence on manual hyperparameter tuning, which limits their automation capability and robustness in complex, multi-scale scenarios. To overcome this limitation, this study proposes a fully automated segmentation framework that synergistically integrates an enhanced U-Net model with a novel hybrid evolutionary optimization strategy. Extensive experiments on public Kaggle and Sentinel-2 datasets demonstrate the superior performance of our method, which achieves a Pixel Accuracy of 96.79% and an F1-Score of 94.75, outperforming various mainstream baseline models by over 10% in key metrics. The framework effectively addresses the class imbalance problem and enhances feature representation without human intervention. This work provides a viable and efficient path toward fully automated remote sensing image analysis, with significant potential for application in large-scale water resource monitoring, dynamic environmental assessment, and emergency disaster management.

## 1. Introduction

High-precision water body extraction holds significant practical value for global water coverage dynamic monitoring, water resource storage assessment, urban water environment management, disaster emergency response, and the protection of aquatic ecologically sensitive areas [1,2]. Particularly in the current context of intensifying global climate change and given the increasing frequency of extreme hydrological events, the development of efficient and reliable water segmentation techniques has become a pressing need. In recent years, driven by breakthrough advances in computer vision and artificial intelligence technologies, water segmentation techniques have undergone a significant technological generational leap [3].

Early research on water body identification and extraction primarily relied on physically prior-driven threshold models [4]. These included rapid segmentation techniques utilizing near-infrared band absorption characteristics [5] or enhanced water indices (e.g., MNDWI, AWEI) [6,7], alongside methods like edge detection operators [8], data mining [9], and object-based approaches [10]. In recent years, deep learning has achieved a technological leap forward through autonomous feature learning and hierarchical modeling [11]. The Fully Convolutional Network (FCN) stands as a milestone in CNN-based semantic segmentation methods [12]. U-Net [13], VGG [14], MobileNet [15], ResNet-50 [16], and Densenet [17] have been extensively applied to water body identification tasks due to their robust feature extraction capabilities and end-to-end learning paradigm. For instance, in 2020, Wang et al. [17] proposed a multidimensional densely connected convolutional neural network for water body identification, which further enhanced model performance. Additionally, Franz et al. [18] systematically compared 32 convolutional neural network architectures, demonstrating that the ResNet50-encoder-based U-Net model achieved optimal performance on their self-constructed river water segmentation dataset.

While the aforementioned models primarily focus on mitigating the impact of spectral reflectance variability (influenced by factors such as weather conditions, solar illumination angles, and land cover types) and noise interference (e.g., clouds, shadows, buildings) to enhance water body identification accuracy, and have achieved certain improvements in precision, their performance remains highly dependent on hyperparameter selection and network architecture design. Manual parameter tuning is not only time-consuming and labor-intensive but also often fails to locate the global optimum, making it difficult to strike an optimal balance between accuracy and computational efficiency. Consequently, model robustness is constrained.

Training deep learning models requires pre-setting certain hyperparameters, such as the network architecture, activation function, and learning rate [19]. These hyperparameter settings directly determine the performance and practical application of deep learning models. Currently, there is no definitive formula for selecting hyperparameters; instead, they rely primarily on a combination of experimentation and experience. This has become a major factor influencing the successful application of deep learning models across various problems. For deep learning models, hyperparameters include weights and hyperparameters. While optimization methods like Adam [20] offer good results for weights, their results are significantly affected by hyperparameters. Improper hyperparameter settings can prevent convergence during model training, leading to suboptimal results. Therefore, efficient hyperparameter optimization algorithms are crucial for deep learning models implemented in remote sensing image interpretation. Our main contributions can be summarized as follows:(1)An evolutionary-hybrid neural framework for automated optimization of water body segmentation models. We develop a novel approach that integrates a multi-algorithm evolutionary optimization system (combining GA, ACO, PSO, and AE) with deep neural networks to automate hyperparameter configuration for water body identification. This framework effectively addresses the critical challenge of manual parameter tuning in deep learning-based water segmentation methods by dynamically optimizing the learning rate, batch size, and momentum, thereby significantly enhancing model performance and training efficiency while reducing human intervention.(2)Adopting a learnable multi-loss function fusion strategy with adaptive weighting, a linearly weighted loss function fusion strategy is proposed to collaboratively optimize classification confidence and geometric consistency. Through adaptive adjustments during training, the contributions of different loss terms to model optimization are balanced, addressing the class imbalance problem in water segmentation. Weights are automatically adjusted during training using an evolutionary algorithm to ensure an optimal balance between the various loss function objectives throughout the learning process.(3)The method was verified on two public remote sensing datasets. It significantly outperformed the baseline model in multiple quantitative indicators (such as mIoU and F1_Score), especially when dealing with complex remote sensing scenes such as urban areas (building shadow interference), cloud cover (noise interference), and multi-scale water bodies (from small rivers to large lakes). It demonstrated stronger generalization ability and boundary segmentation accuracy, providing a better automated solution for the automated high-precision extraction of remote sensing water bodies.

## 2. Related Work

In this section, first, we first review related work from two perspectives: deep learning-based water body interpretation methods and deep model parameter optimization using evolutionary strategies.

### 2.1. Water Body Interpretation Method Based on Deep Learning Model

Deep learning primarily utilizes artificial neural networks to autonomously learn and extract features from remote sensing imagery, thereby avoiding potential human bias caused by manual feature selection and reducing reliance on prior knowledge. Deep learning demonstrates high accuracy and robustness when processing remote sensing imagery with complex backgrounds and diverse textures [21]. For example, Chen Qian et al. [22] used the DeepLabv3 semantic segmentation model to isolate water features from high-resolution satellite imagery, mitigating the effects of shadows and buildings more effectively than traditional classification and machine learning techniques. The fully convolutional neural network (FCN) and its variants, such as UNet, SegNet, and DeepLab, have become the mainstream deep learning semantic segmentation models for water extraction and are widely used in water extraction from remote sensing imagery [23,24]. The integration of deep learning and remote sensing technology has made new avenues for surface water extraction, water environment monitoring, and the study of global hydrological phenomena. The above methods have achieved remote sensing water body segmentation to a certain extent, but most of these methods require manual adjustment of classifier hyperparameters.

### 2.2. Deep Model Parameter Optimization

Evolutionary Algorithms (EAs) [25,26], which are global optimization methods based on the principles of biological evolution, have been introduced to optimize deep learning models. By simulating biological processes such as natural selection, crossover, and mutation, these evolutionary algorithms can efficiently search for optimal solutions in complex parameter spaces. Common evolutionary algorithms include the Genetic Algorithm (GA) [27], Ant Colony Optimization (ACO) [28,29], and Particle Swarm Optimization (PSO) [30]. These frameworks achieve high-precision performance while focusing on automated model optimization, efficient architecture discovery, and robust training strategies, significantly reducing reliance on manual intervention. Li et al. [31] introduced a new evolutionary computation framework aided by machine learning, named EVOLER, which for the first time enables the theoretically guaranteed global optimization of complex non-convex problems; Liang et al. [32] proposed a multi-objective evolutionary NAS frame-work based on a weight-sharing supernet to improve the search efficiency of traditional evolutionary-computation-based NAS; Zhang et al. [33] progressed a remote sensing scene classification with a parameter learning algorithm-based co-evolutionary. Esteban et al. [34] proposed a parallel evolutionary algorithm for image classification, which optimized both network architectures and hyperparameters, demonstrating significant improvements in classification accuracy. Furthermore, in 2023, Abdullah et al. [35] introduced an Evolutionary Deep Convolutional Neural Network System (EDCNNS) for Alzheimer’s disease (AD) diagnostic systems, achieving substantial reduction in computational latency while enhancing diagnostic accuracy.

## 3. Proposed Methods

### 3.1. Overall Framework

The overall framework of the proposed water body identification is illustrated in Figure 1. It mainly consists of three main modules: (1) Hyperparameter optimization: evolutionary algorithms is used to optimize model hyperparameters, generating the optimal settings through a hybrid strategy. (2)Model training: An encoder–decoder architecture is used to train the water segmentation model. (3) Evaluation: A weighted loss function combining classification loss and Dice Loss is used for evaluation.

During image preprocessing, the model conducts data augmentation and normalization. The normalization scales the input image pixel values from [0, 255] to the range [0, 1] in order to accelerate model convergence and training speed. Data augmentation is performed via random scaling, flipping, and color transformation to enhance data diversity and improve the model’s generalization capability. Once preprocessing is complete, the processed image data is converted into PyTorch 2.3.0 tensors and transferred to the GPU. The bundled tensors are then fed into the U-Net model for training.

Prior to training the model, it is necessary to use an evolutionary algorithm to search for the best combination of hyperparameters. A search space containing the hyperparameters to be optimized, that is, a hyperparameter pool, is constructed manually. This pool includes key hyperparameters that significantly affect model performance, such as learning rate, batch size, and optimizer type. A combination of evolutionary algorithms is then selected. During the search process, the algorithm continuously generates new hyperparameter combinations according to a predetermined strategy. For each new hyperparameter combination, the evolutionary algorithm activates a fitness evaluation mechanism. This evaluation is performed by applying the hyperparameter combination to the U-Net model and conducting training and validation on a designated validation dataset. The model’s performance on the validation dataset, such as classification accuracy, segmentation precision, and convergence speed, is quantified as a fitness value. A higher fitness value indicates that the hyperparameter combination enables the model to perform better in the current task. Following the survival-of-the-fittest principle, the evolutionary algorithm selects hyperparameter combinations with excellent performance until a termination condition is met, finally outputting the optimal configuration.

### 3.2. Backbone

#### 3.2.1. Network Model Selection

Common neural network models used for image identification include basic convolutional neural networks, deep training optimized networks, and attention enhancement networks [36]. A classical convolutional neural network (CNN) [37] suffers from the loss of spatial information due to pooling operations; the absence of a contextual information fusion mechanism renders it insufficiently sensitive to complex backgrounds such as the water and land boundary; and single-scale feature extraction proves inadequate for adapting to multi-scale objects [38]. The traditional deep training optimization network ResNet lacks channel attention and does not explicitly model inter-channel relationships [39], which may lead to the neglect of important spectral features; moreover, it is not suitable for pixel-level tasks because an effective upsampling mechanism is missing when it is directly employed for segmentation. The classical attention enhancement network SENet mainly focuses on channel weighting without explicitly optimizing spatial features [40], which potentially causes the loss of edge details; in addition, the absence of a multi-scale feature fusion mechanism makes it difficult to adapt to complex scenes.

In contrast to the aforementioned networks, the improved U-Net proposed in this work adopts a backbone structure that employs ResNet50 as the encoder. The use of residual structures alleviates gradient vanishing and facilitates the extraction of deeper and more discriminative features. This enhancement improves the capability to capture complex textures such as water ripples and turbid regions. Simultaneously the improved U-Net introduces an attention optimization mechanism by integrating the CBAM module and designing a dynamic gating fusion unit to achieve the synergistic optimization of spectral and spatial features. Furthermore, the improved U-Net employs a loss function optimization strategy. In response to the significant disparities in water body proportions typical of remote sensing imagery, a weighted multi-loss function is constructed. By jointly applying cross entropy loss, Dice Loss, and Focal Loss, the class imbalance are alleviated and the overall robustness of the model is improved.

#### 3.2.2. The Network Structure

Traditional U-Net architectures generally employ simple convolution layers and pooling layers in the encoder, resulting in relatively simple structures. In the improved U-Net model, a more complex pre-trained network, ResNet50, is selected as the backbone. The output feature maps from ResNet50 are utilized in the decoder. By fusing these via skip connections with the decoder’s feature maps, high-resolution feature information is recovered, and water body features are accurately recognized. In the U-Net model, ResNet50 serves as the encoder responsible for extracting features from the input image. Specifically, feature maps extracted at different levels of ResNet50 are transmitted to the decoder via skip connections. This design not only preserves the strong feature extraction capability of ResNet50 but also achieves high-precision pixel-level segmentation through the U-Net decoder structure.

ResNet50 extracts multi-scale feature information from the input image using its residual modules. Each residual module is described by (1):(1)$$\mathrm{ResidualBlock}(x)\mathrm{ReLU}(x+\mathrm{Con}v_{3}(\mathrm{ReLU}(\mathrm{Con}v_{2}(\mathrm{ReLU}(\mathrm{Con}v_{1}\left(x)))))\right)$$where three convolution layers (Conv_1_ Conv_2_, and Conv_3_) are applied sequentially with the ReLU activation function, a skip connection, denoted as x, bypasses the convolutional layers and directly adds the input to the output, facilitating gradient flow and improving training efficiency.

In the decoder of U-Net, the feature maps extracted by ResNet50 are fused with the decoder feature maps via skip connections. This skip connection is implemented as shown in (2):(2)$$\mathrm{SkipConnection}\left(x,y\right)=\mathrm{Concat}\left(\mathrm{UpSample}\left(x\right),y\right)$$
where x represents the feature map extracted by ResNet50, y is the decoder feature map, Upsample indicates the upsampling operation, and Concat the concatenation operation. In the decoder, the fused feature maps are further processed by convolution layers and activation functions for feature fusion. This fusion is expressed in (3):(3)$$\mathrm{FeatureFusion}\left(x\right)=\mathrm{ReLU}\left(\mathrm{Conv}\left(\mathrm{ReLU}\left(\mathrm{Conv}\left(x\right)\right)\right)\right)$$
where Conv signifies a convolution layer and ReLU denotes an activation function.

#### 3.2.3. Dynamic Parameter Adjustment

We introduce the Dynamic Gate mechanism to achieve adaptive adjustment of attention weights through feature complexity sensing. This module is embedded in the enhanced CBAM (Enhanced CBAM) and integrated into the feature fusion layer (unetUp module) of the U-Net decoder. The overall architecture comprises three core components:Spectral Attention Layer: Uses multi-scale one-dimensional convolution to capture cross-channel spectral dependency.Spatial Attention Layer: Based on direction-aware convolution to extract spatial contextual feature.Dynamic Gating Layer: Generates spatially adaptive attention intensity and feature retention coefficients.

In the dynamic adjustment mechanism, we assume the input feature is $X\in R^{B\times C\times H\times W}$. The dynamic parameter adjustment process follows the mathematical principles below.

First, the spectral weight calculation is performed. After compressing the spatial dimensions using global average pooling, cascaded 1D convolutional layers (with kernel sizes of 5 and 3, respectively) are applied to extract cross-channel spectral dependencies. The large convolution kernel captures global spectral trends, while the small kernel focuses on local band interactions. Its mathematical expression is given as (4). After this operation, the model automatically enhances the weights for water-sensitive bands such as near-infrared (NIR) and suppresses interference from visible light bands in shadow regions. The model’s noise robustness is further enhanced by smoothing out abnormal reflectance fluctuations induced by sensor noise(4)$$W_{s}=s\left(f_{3\times 1}\left(d\left(f_{5\times 1}\left(\mathrm{GAP}(X)\right)\right)\right)\right)$$

For the second step, spatial weight generation, it is necessary to fuse average pooling (to retain the overall distribution) and max pooling (to highlight salient features). Orthogonally separated 7×1 and 1×7 convolution kernels are used to extract spatial context in the horizontal and vertical directions, respectively:(5)$$W_{p}=\sigma \left(f_{7_{v}}\left(f_{7_{h}}\left[\mathrm{AvgPool}(X);\mathrm{MaxPool}(X)\right]\right)\right)$$

At this stage, the horizontal convolution kernel reinforces the continuity of linear targets such as rivers and canals, while the vertical convolution kernel enhances the contrast between lake boundaries and land, thereby strengthening directional awareness. Pooling operations couple features from different receptive fields, enhancing the aggregated representation of fragmented water bodies such as pond clusters.

Finally, during the dynamic mixing output, spatially adaptive adjustment coefficients, including attention intensity (γ) and feature retention rate (β), are generated based on feature complexity. These coefficients are obtained through a lightweight network, as shown in (6):(6)$$[\gamma ,\beta ]=\Gamma_{\mathrm{complexity}}\left(x\right)$$
where $\Gamma_{\mathrm{complexity}}$ is constructed by global average pooling followed by two convolution layers with a compression ratio of 16:1, and the final output complies with a specific constraint. The feature fusion is then formulated as in (7):(7)$$X_{\mathrm{out}}=\gamma ⊙(W_{p}⊙X)+\beta ⊙X$$

After the dynamic mixed output operation, in regions with high complexity, the attention intensity γ increases and the feature retention rate β decreases, thereby enhancing the characterization of boundary features by the attention mechanism γ. In contrast, in relatively simple regions, the attention intensity β decreases and the feature retention rate increases, which helps preserve low-level spectral features and prevents over-smoothing. The design of the residual connection ensures that gradients can directly bypass to the original features, mitigating gradient attenuation problems in deep networks. The process of the dynamic parameter adjustment mechanism described above is illustrated in Figure 2.

### 3.3. Hyperparameter Tuning Combined with Multiple Evolutionary Algorithms

To overcome the inherent trade-off between global exploration and local convergence in single-method optimization, this study introduces an integrated hyperparameter optimization framework combining Genetic Algorithm (GA), Ant Colony Optimization (ACO), Particle Swarm Optimization (PSO), and the Alpha Evolutionary (AE) algorithm. Each algorithm is assigned a distinct role according to its operational principles: GA performs wide-ranging exploration through selection, crossover, and mutation to preserve population diversity; ACO employs a pheromone-mediated positive feedback mechanism to intensify search trajectories in promising regions; PSO accelerates convergence by steering particles based on individual and collective experience; and AE concentrates on refining the elite solution space during later stages to secure stable, high-precision outcomes. Orchestrating their collaboration, a dynamic weight fusion mechanism adaptively modulates the contribution of each algorithm in response to iterative progress and performance feedback, thereby harmonizing global exploration, path refinement, rapid convergence, and solution-space polishing to achieve fully automated and near-optimal hyperparameter configuration.

#### 3.3.1. Hyperparameter Optimization

The hyperparameter optimization in this work refers to the optimization of the learning rate, batch size, and momentum for the improved U-Net model. After initializing the hyperparameter values using an evolutionary algorithm, the hyperparameter array is input into the U-Net model for prediction. The search terminates after a specified number of iterations. All evaluation results of water body identification are then ranked from best to worst, and the hyperparameter array with the optimal evaluation result is selected for subsequent formal water body identification.

In this study, the evaluation function used in the hyperparameter optimization module is the F1_Score.(8)$$F_{1}\_\mathrm{Score}=2\cdot \frac{\mathrm{Precision}\cdot \mathrm{Recall}}{\mathrm{Precision}+\mathrm{Recall}}$$(9)$$\mathrm{Precision}=\frac{\mathrm{TP}}{\mathrm{TP}+\mathrm{FP}},\mathrm{Recall}=\frac{\mathrm{TP}}{\mathrm{TP}+\mathrm{FN}}$$(10)$${\left(x_{\mathrm{LearningRate}},x_{\mathrm{BatchSize}},x_{\mathrm{Momentum}}\right)}^{*}=\mathrm{argmax}\;f_{F1\_\mathrm{Score}}\left(x_{\mathrm{LearningRate}},x_{\mathrm{BatchSize}},x_{\mathrm{Momentum}}\right)\;x_{\mathrm{LearningRate}}\in X_{\mathrm{LearningRate}},x_{\mathrm{BatchSize}}\in X_{\mathrm{BatchSize}},x_{\mathrm{Momentum}}\in X_{\mathrm{Momentum}}$$

Its mathematical expression is given in (8), and the associated parameter expressions are provided in (9). Therefore, the hyperparameter optimization problem is defined by the mathematical model expressed in (10), where ${\left(x_{\mathrm{LearningRate}},x_{\mathrm{BatchSize}},x_{\mathrm{Momentum}}\right)}^{*}$ represents the hyperparameter configuration that produces the optimal value $f_{F1\_\mathrm{Score}}$. The hyperparameter set $\left(x_{\mathrm{LearningRate}},x_{\mathrm{BatchSize}},x_{\mathrm{Momentum}}\right)$ can take any value from the search space $\left(X_{\mathrm{LearningRate}},X_{\mathrm{BatchSize}},X_{\mathrm{Momentum}}\right)$.

#### 3.3.2. Weight Adaptive Optimization Based on Multi-Algorithm Fusion

In view of the tendency of a single evolutionary algorithm to fall into local optima and its limited convergence speed, this work proposes a four-stage joint optimization strategy that integrates Genetic Algorithm (GA), Ant Colony Optimization (ACO), Particle Swarm Optimization (PSO), and Alpha Evolutionary Algorithm (AE) [41]. Through the construction of a dynamic weight fusion mechanism and a layered optimization framework, this strategy combines the dual advantages of global search and local refinement to achieve collaborative exploration of the hyperparameter space as well as fine adjustment in the solution space.

The core innovations of this strategy are reflected in the following four aspects:

Multi-Algorithm Joint Prediction Model: Assume that the outputs of the U-Net model optimized by GA, ACO, PSO, and AE are $P_{\mathrm{GA}}$, $P_{\mathrm{ACO}}$, $P_{\mathrm{PSO}}$, and $P_{\mathrm{AE}}$, respectively. The joint prediction result is generated by weighted fusion as expressed in (11):(11)$$P_{\mathrm{Ensemble}}=\omega_{\mathrm{GA}}\times P_{\mathrm{GA}}+\omega_{\mathrm{ACO}}\times P_{\mathrm{ACO}}+\omega_{\mathrm{PSO}}\times P_{\mathrm{PSO}}+\omega_{\mathrm{AE}}\times P_{\mathrm{AE}}$$
where $\sum \omega_{i}=1$, the weight coefficients are dynamically adjusted. This model integrates the population diversity of GA, the positive feedback path optimization of ACO, the gradient-driven characteristics of PSO, and the advanced comprehensive optimization capability of AE, thereby forming a balanced architecture of global exploration and local refinement.

The dynamic weight update mechanism significantly impacts the final performance of the model, making the fine-tuning of joint weights a critical step. By dynamically adjusting the weights to achieve optimal results, the process can be divided into three stages. In the early iteration stage (*T*1), the weight of the GA (Genetic Algorithm) is increased to approximately 0.35 ± 0.05, which enhances the global topological search capability and prevents premature convergence. In the middle iteration stage (*T*2), a balance is dynamically achieved between the ACO (Ant Colony Optimization) weight, set to approximately 0.28, and the PSO (Particle Swarm Optimization) weight, set to approximately 0.32. This forms a dual driving mechanism based on path optimization (via ACO) and gradient-driven refinement (via PSO), enabling efficient exploration and exploitation of the solution space. Finally, in the late-convergence stage (*T*3), the AE (Autoencoder) weight is increased to approximately 0.4 ± 0.06, activating the elite solution space compression mechanism to refine and converge on the optimal result. Table 1 descripts their individual functions.

### 3.4. Loss Function

To jointly optimize classification confidence and geometric consistency, this work proposes a linear weighted loss fusion strategy. It combines the classification loss (either Cross-Entropy Loss or Focal Loss) with Dice Loss to form a composite loss function as shown in (12):(12)$$L_{\mathrm{Total}}=\lambda \times L_{\mathrm{cls}}+\beta \times L_{\mathrm{dice}}$$
where $L_{\mathrm{cls}}$ denotes the classification loss (either $L_{\mathrm{CE}}$ or $L_{\mathrm{Focal}}$), $L_{\mathrm{Dice}}$ represents Dice Loss, and λ and β are dynamic weighting coefficients. The default settings are λ=1 and β=1. These coefficients can be adaptively adjusted during training using evolutionary algorithms (such as Genetic Algorithm or Particle Swarm Optimization) to balance the contributions of the different loss components in model optimization. This strategy, through end-to-end joint training, enables simultaneous optimization of the following objectives during backpropagation:

Optimization of Classification Confidence: Constraining the model output probability distribution to match the true labels to enhance pixel-level classification accuracy.

Optimization of Geometric Consistency: Improving the topological alignment between the predicted regions and the ground truth mask to refine boundary segmentation performance.

## 4. Experiments and Results

### 4.1. Datasets

The datasets were obtained from two sources. One dataset was published on Kaggle.com by Francisco Escobar for training water body identification models. In this dataset each image is provided with a black-and-white mask in which white represents water and black represents all non-water areas. These masks are generated through the computation of NDWI (Normalized Water Difference Index). NDWI is frequently used to detect and measure vegetation in satellite imagery but a higher threshold is employed for water detection.

The second dataset consists of atmospherically corrected multispectral Sentinel-2 images [40]. The Sentinel-2 dataset provides continuous coverage for global land surface monitoring and offers several advantages compared with other image types such as high spatial resolution and free data access. Because the input to the model must conform to the VOC format, the masks are required to set all non-target objects to a pixel value of 0 and the target objects to a pixel value of 1. To meet this requirement, the masks in this dataset are converted to the corresponding format through a script, thereby ensuring that the model correctly identifies water body areas and performs water body identification with higher accuracy.

During preprocessing, we used the Kaggle dataset as supplemental training data for the Sentinel-2 dataset to prevent model overfitting and improve generalization and stability. The primary dataset, which served as the basis for model training and evaluation, was manually digitized to generate ground truth. For areas where visual assessment was challenging, we referred to auxiliary high-resolution imagery from sources like Google Earth for accurate fine-scale identification. The two datasets were merged during the pre-processing phase after being split into 512 × 512 patches, which served as the model input. The combined dataset was then partitioned into 70% for training, 15% for validation, and 15% for testing.

### 4.2. Evaluation Metrics

This study employs five evaluation metrics to assess the experimental results. These metrics include Pixel Accuracy (PA), F1_Score, Mean Intersection-over-Union (mIoU), Mean Pixel Accuracy (mPA), and Mean Precision (mPrecision).

Pixel Accuracy (PA) reflects the overall capability of the model to distinguish water pixels from non-water pixels. Its value ranges from 0 to 1; a value closer to 1 indicates a stronger ability to correctly classify true water bodies. The mathematical expression for PA is given in (13).(13)$$\mathrm{PA}=\frac{\mathrm{TP}+\mathrm{TN}}{\mathrm{TP}+\mathrm{TN}+\mathrm{FP}+\mathrm{FN}}$$

F1_Score evaluates the comprehensive performance of the model on imbalanced datasets. Its value ranges from 0 to 1. F1 is the harmonic mean of precision and recall; a value closer to 1 signifies superior performance on imbalanced data. The mathematical expression for the F1_Score provided in above (8).

Mean Intersection-over-Union (mIoU) quantifies the geometric consistency between the predicted regions and the true regions. The value ranges from 0 to 1; a higher mIoU signifies better overlap between prediction and truth and indicates more precise boundary segmentation. The mathematical expression for mIoU is given in (14):(14)$$\mathrm{mIoU}=\frac{1}{C}\sum_{c=1}^{C}\frac{TP_{c}}{TP_{c}+FP_{c}+FN_{c}}$$
where C represents the class and two classes are used (water and background). Mean Pixel Accuracy (mPA) measures the classification accuracy of each class independently. Its value ranges from 0 to 1; a higher mPA indicates that the model exhibits stronger robustness against local noise and provides better suppression of false detections in localized areas. The mathematical expression for mPA is shown in (15).(15)$$\mathrm{mPA}=\frac{1}{C}\sum_{c=1}^{C}\frac{TP_{c}}{TP_{c}+FN_{c}}$$

Mean Precision (mPrecision) assesses the reliability of positive predictions. Its value ranges from 0 to 1; a higher mPrecision indicates that a larger proportion of samples predicted as positive are indeed positive, implying a lower false alarm rate. The mathematical expression for mPrecision is provided in (16).(16)$$\mathrm{mPrecision}=\frac{1}{C}\sum_{c=1}^{C}\frac{TP_{c}}{TP_{c}+FP_{c}}$$

Collectively these evaluation metrics assess the performance of the water body identification model from multiple perspectives. PA and F1_Score validate overall global performance, whereas mIoU and mPA provide a dual evaluation of global geometric accuracy and local classification accuracy, and mPrecision primarily constrains the false positive rate.

### 4.3. Results and Analysis

Baseline models used for comparison include U-Net variants that incorporate different backbone networks. In this work the backbone networks comprise VGG16, DenseNet, and MobileNet. Two optimizers, namely the SGD optimizer and the Adam optimizer, are also compared.

To objectively compare the prediction results of the improved U-Net model with those of the baseline model the hyperparameters for the baseline models are set to be identical to those of the experimental model. These parameters include the learning rate, optimizer type, batch size, and the number of training epochs. Pre-trained weights are used to initialize the backbone network for each model. All network models paired with each backbone network are integrated within the U-Net architecture. All eight backbone-optimizer combinations, ResNet50 + Adam, ResNet50 + SGD, VGG16 + Adam, VGG16 + SGD, DenseNet + Adam, DenseNet + SGD, MobileNet + Adam, and MobileNet + SGD, used identical hyperparameters: a maximum learning rate of 1 × 10^−4^, minimum learning rate of 1 × 10^−6^, and 200 training batches.

To investigate the superiority of the proposed model, eight combinations formed by four different backbone networks and two optimizers are compared using the five-evaluation metrics mentioned above. The evaluation results are presented in Figure 3 and Table 2.

It is observed that the combination of the ResNet backbone with the Adam optimizer exhibits a more stable training process compared with the other seven baseline models. During the iterative process both PA and F1_Score remain higher than those obtained from the other seven models. After the final iteration the best PA and best F1_Score are significantly higher than those from the other baseline models. With the exception of the ResNet combined with the SGD optimizer the ResNet + Adam combination outperforms the other models by more than 10 percent. This demonstrates the superiority of the ResNet + Adam combination, and the visualization results can be seen in Figure 4, where we can intuitively observe that the water body extraction result obtained by the combination of ResNet + Adam is closest to the original image and the ground truth, especially in water body edge details and small water bodies.

### 4.4. Ablation Experiment

Ablation experiments demonstrate the effects of evolutionary algorithms, attention modules, linear weighted loss function fusion, and evolutionary algorithms on water extraction from remote sensing images. It can be observed that after incorporating the CBAM attention mechanism the model attains a PA value exceeding 0.965, which is well above the acceptable threshold of 0.90. The F1_Score exceeds 0.945, surpassing the minimal acceptable value of 0.85. Similarly, mIoU reaches above 0.902, mPA exceeds 94.4, and mPrecision reaches above 0.952, all of which are within excellent ranges. Compared with the U-Net model without the CBAM attention mechanism, the network with CBAM consistently outperforms across all evaluation metrics and exhibits a more stable training process. This demonstrates the effectiveness of the proposed strategy. To assess the effectiveness of loss function fusion for enhancing water body identification performance the model employing the unfused cross entropy loss and Focal loss is compared with the model using the fused Focal + Dice loss function. The comparison results are visualized, and evaluation results are provided in Table 3.

After fusing the loss functions the model attains a PA exceeding 0.965, an F1_Score above 0.945, an mIoU above 0.902, an mPA exceeding 0.944, and an mPrecision above 0.952. Apart from the mPrecision metric the network employing the fused loss functions outperforms the networks using only cross entropy loss or Focal loss. This further verifies the effectiveness of the proposed strategy.

To examine the effectiveness of the joint evolutionary algorithm for hyperparameter optimization, the evaluation metrics obtained from four individual evolutionary algorithms are compared with those achieved using the joint evolutionary algorithm. The results are shown in Table 4 and visualized as shown in Figure 5.

It indicates that the model optimized by the joint algorithm outperforms the models optimized by any single evolutionary algorithm in all evaluation metrics. Furthermore, the evaluation results demonstrate that the Alpha Evolutionary Algorithm yields superior performance compared with the other three evolutionary algorithms. This indirectly confirms that assigning a higher weight to the Alpha Evolutionary Algorithm in the joint weighting strategy is appropriate. The water body comparison results are shown in Figure 6.

## 5. Conclusions

This study addressed the challenges of automating water body identification from satellite imagery by introducing a U-Net model optimized through a hybrid evolutionary algorithm. The core of our approach lies in the integration of three key components: (1) an enhanced U-Net with a ResNet50 encoder and a dynamic attention mechanism for improved multi-scale feature extraction and adaptive feature calibration; (2) a hybrid evolutionary strategy combining the Genetic Algorithm, Ant Colony Optimization, Particle Swarm Optimization, and the Alpha Evolutionary algorithm to automate hyperparameter configuration, mitigating the reliance on manual tuning; and (3) a learnable multi-loss fusion strategy to balance classification and geometric consistency, effectively addressing class imbalance. Extensive experiments on the Kaggle and Sentinel-2 datasets demonstrated the efficacy of the proposed method. Our model achieved a Pixel Accuracy of 96.79% and an F1-Score of 0.9475, outperforming various baseline models by a considerable margin (over 10% in key metrics). Ablation studies confirmed the contribution of each component, showing that the hybrid evolutionary optimizer yielded better performance than any single algorithm alone.

Despite these results, we note two primary limitations. First, the model’s training relied on publicly available Sentinel-2 imagery, and its performance on data from other sensors or under significantly different imaging conditions requires further validation. Second, the joint optimization of multiple evolutionary algorithms is computationally intensive, leading to increased training time and resource demands. Future work will focus on several directions to enhance the framework’s practicality and scope. This includes incorporating multi-source remote sensing data (e.g., Landsat, GF series, SAR) to improve generalization, exploring lightweight evolutionary strategies or surrogate models to reduce computational costs, and extending the model’s capability to multi-class land-cover segmentation and temporal analysis for dynamic monitoring. The research framework presented here offers a viable path toward fully automated remote sensing image analysis, with potential applications in water resource management, environmental monitoring, and disaster assessment.

## Figures and Tables

**Figure 1 biomimetics-10-00732-f001:**
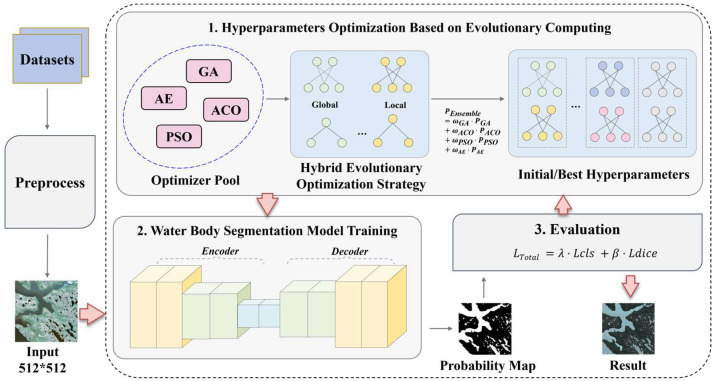
The water body identification framework based on hybrid evolutionary algorithms.

**Figure 2 biomimetics-10-00732-f002:**
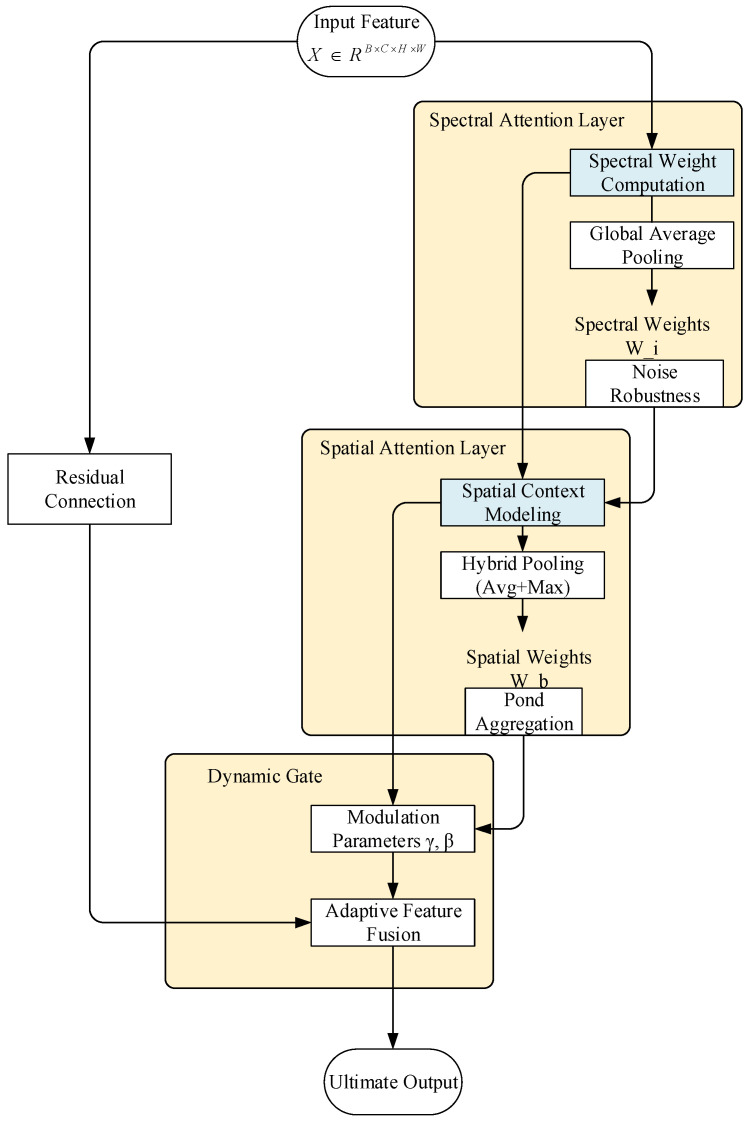
Dynamic Parameter Adjustment.

**Figure 3 biomimetics-10-00732-f003:**
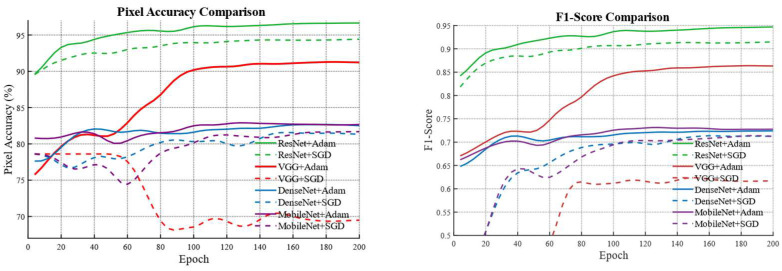
Comparison between the U-Net model with the Adam optimizer and other baseline models.

**Figure 4 biomimetics-10-00732-f004:**
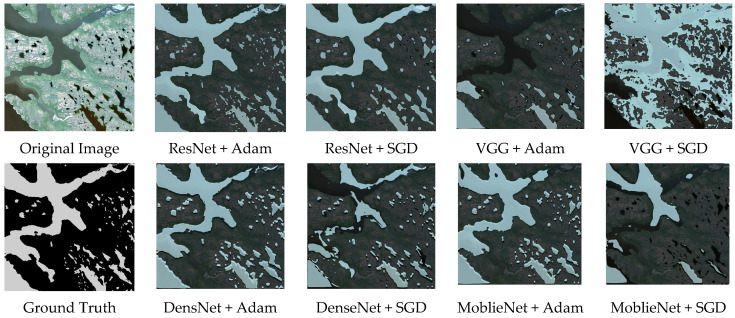
Comparison of water extraction results with different backbone network and optimizer combinations.

**Figure 5 biomimetics-10-00732-f005:**
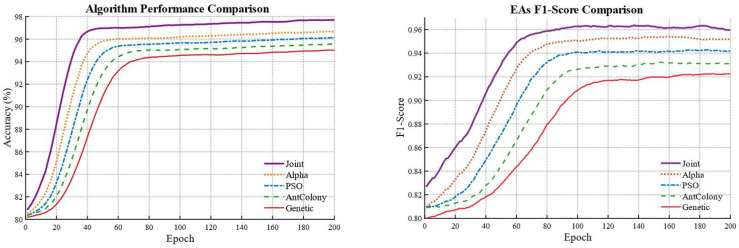
Comparison of Evaluation Metrics for Evolutionary Algorithms and the Combined Algorithm.

**Figure 6 biomimetics-10-00732-f006:**
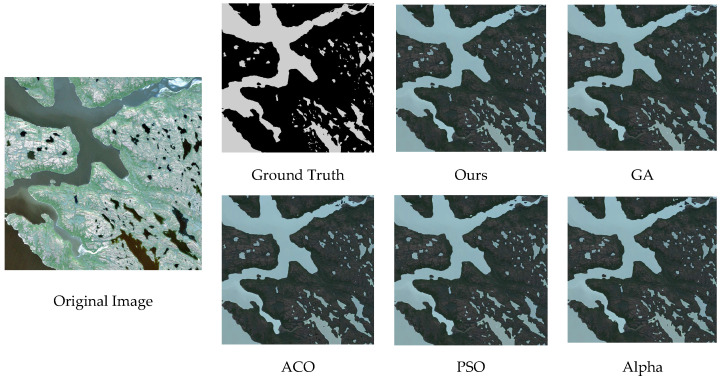
Comparison of water body segmentation results with different evolutionary algorithm.

**Table 1 biomimetics-10-00732-t001:** Parameters and tasks involved in each evolutionary algorithm.

Algorithm	Parameter and Task	Initial Value
GA	Population Size: Determines the diversity of potential hyperparameter sets in the gene pool.	20
	Crossover Rate: Controls the probability of combining genetic material from two parents to produce offspring.	0.7
	Mutation Rate: Introduces random changes to offspring, maintaining population diversity and preventing premature convergence to local optima.	0.1
	Selection Method: Governs which individuals are chosen for reproduction. Tournament selection provides good selective pressure.	Tournament (size = 3)
ACO	Number of Ants: Analogous to population size. Each ant constructs a solution (a path representing a hyperparameter set).	20
	Pheromone Influence (α): Determines the weight of pheromone trails in path selection. Higher values increase exploitation of known good paths.	1.0
	Heuristic Influence (β): Determines the weight of heuristic information in path selection. Higher values favor exploration.	2.0
	Pheromone Evaporation Rate (ρ): Simulates the evaporation of pheromones over time, preventing unlimited accumulation and allowing the colony to forget poorer paths, facilitating exploration.	0.5
PSO	Number of Particles Similarly to population size. Each particle represents a potential hyperparameter set flying through the search space.	20
	Inertia Weight (w): Balances exploration (high w) and exploitation (low w). It controls the particle’s momentum based on its previous velocity.	0.5
	Cognitive Coefficient (c1): Attracts the particle towards its personal best position (pbest), encouraging exploitation of personally found good solutions.	1.5
	Social Coefficient (c2): Attracts the particle towards the swarm’s global best position (gbest), encouraging convergence towards the best-known solution.	2.0
AE	Population Size: Determines the number of candidate solutions maintained per iteration.	20
	Decay Coefficient of Disturbance: Controls the decaying speed of the disturbance intensity (alpha) during the algorithm’s iterative process	0.9
	Evaporation Rate: Simulates the pheromone evaporation mechanism like Ant Colony Optimization	0.2

**Table 2 biomimetics-10-00732-t002:** Maximum Pixel Accuracy and F1_score For Each Backbone and Optimizer Combination.

Backbone + Optimizer	Best PA(%)	Best F1_Score(%)
ResNet50 + Adam	**96.7897**	**94.75**
ResNet50 + SGD	94.4729	91.59
VGG16 + Adam	81.2445	69.24
VGG16 + SGD	78.6100	65.83
DenseNet + Adam	82.8770	72.59
DenseNet + SGD	81.8651	71.67
MobileNet + Adam	83.3628	73.45
MobileNet + SGD	81.9363	71.78

**Table 3 biomimetics-10-00732-t003:** Analysis of the Ablation Effects of Individual Components.

Methods	PA(%)	F1_Score(%)	mIoU(%)	mPA(%)	mPrecision(%)
U-Net	96.28	93.90	88.87	93.84	94.66
U-Net +CELoss	96.67	94.50	89.67	93.85	95.49
U-Net +FocalLoss	96.70	94.50	89.91	93.90	95.46
U-Net +FocalLoss + Dice Loss + CBAM	96.93	94.90	90.61	94.62	95.57

**Table 4 biomimetics-10-00732-t004:** Maximum Pixel Accuracy and F1_score For evolutionary algorithms.

Algorithm	Best PA(%)	Best F1Score(%)
Genetic Algorithm (GA)	95.58	92.21
Ant Colony Optimization (ACO)	95.79	93.15
Particle Swarm Optimization (PSO)	96.11	94.31
Alpha Evolutionary (AE)	96.53	95.57
The Joint: GA + ACO + PSO + AE	**97.69**	**96.32**

## Data Availability

The study did not report any data.

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
