# Peer review of "Water Body Identification from Satellite Images Using a Hybrid Evolutionary Algorithm-Optimized U-Net Framework"

_biomimetics, 2025, doi:10.3390/biomimetics10110732_

Round 1

Reviewer 1 Report

Comments and Suggestions for Authors

Manuscript Title: Water Body Identification using Neural Network Model Optimized by Evolutionary Algorithms from Satellite Images.

Reviewer’s Comment:

  1. The abstract presented in the manuscript is presents the problem, objectives, methodology and contribution however, some of the other components that should be there in the abstract is missing, for instance: mention dataset employed, quantitative performance results with a clear comparative statement that highlights how the proposed approach outperforms the traditional methods. Kindle reframe the abstract based on the same.

  1. Check Figure 1: The water body identification model performing the water body identification task, the proposed framework does not at all illustrate the underlying process clearly. You should recreate Figure 1 that would be able to provide a better understanding and the detailed proposed framework including all the essential components. Also, check the spelling of “input image” written as “Iuput”.

  1. The Figure number needs to be checked thoroughly. for instance: Figure 8: Comparison of water body segmentation results. (Figure 4, 6 and 7) seems missing. Also, the numbering of Table should be thoroughly checked. Table No 2 is Refer to Table 3: Analysis of the Ablation Effects of Individual Components which should be Table 2.

  1. Figure 8: The comparison of water body segmentation results in the manuscript is placed without sufficient context or explanation. No where has the author specified what the segmentation results represent, which algorithms or optimization techniques were applied, or what is the threshold values and what is number of thresholds considered for the segmentation. The absence of such description makes it very tough for the reader to understand as well as interpret the figure accurately.

  1. In addition to comment number 4 on Figure 8, the number of visual representations depicted for the comparison seem inadequate, making it very difficult to roughly evaluate the differences in segmentation performance across the methods considered.

  1. Kindly refer to Table 1: Maximum Pixel Accuracy and F1-score For Each Backbone and Optimizer Combination. For the backbone + optimizer (VGG16+Adam) the values for F1 Score and Maximum Pixel Accuracy is left blank. What does that indicate? Does that mean the experiment could not run / perform due to computational limit or any resource constraints, or does that mean the model failed to converge or any other reason. Leaving it blank without clarification is not acceptable and raises several questions. Kindly provide the reason.

On the contrary, the Figure 3, that portrays the same information in visual representation has depicted the PA and F1-score generated by each Backbone and Optimizer Combination even for VGG16+Adam which was left blank in Table 1. That totally makes the entire scenario confusing.

  1. What is the major intention behind combining all these optimization techniques namely, Genetic Algorithm (GA), Ant Colony Optimization (ACO), Particle Swarm Optimization (PSO), and Alpha Evolutionary (AE) algorithm? Furthermore, the manuscript does not clearly elaborate and highlight how each of the optimization techniques contributes in automating hyperparameter configuration and what is their individual roles in the proposed framework. Clarification required.

  1. In addition to comment number 7, the manuscript though makes use of multiple optimization algorithms. However, it would benefit from a clear portrayal of the parameters chosen for each algorithm, their intended purpose, and the initialization values employed. Including such information would definitely enhance not just the transparency, but also the overall impact of the work.

  1. Please refer to Figure 5: Comparison of Evaluation Metrics for Evolutionary Algorithms and the Combined Algorithm (Proposed). The tabular representation of the performance parameters achieved by the proposed and the compared algorithm needs to be highlighted. In addition, it has to be clearly mentioned which dataset has been considered to evaluate the outcome to ensure clarity of the results.

  1. The paper needs to be carefully read once. There are many typos and grammatical errors in the manuscript. Several issues related to Figure and Table numbering exist. The major weakness of the paper is the limited additional knowledge given here. Kindly make the necessary changes.

Author Response

Please refer to the attached document for more details.

Reviewer 2 Report

Comments and Suggestions for Authors

The research work tracked on the Water Body Identification through the satellite images using Neural network model optimized by GA,Ant ACO, PSO, and AE algorithms. The work is more interesting and contain good novelty.  The results of different deep learning models tuned by EA algorithms are reliable. The comments are:
1. In title , the authors remove the word 'neural network model'. Revise the title and to be focused towards the content of the manuscript.
2.The first 2 paragraphs must cite with  the appropriate references in 3.2.1 Network Model Selection.This will ensure the weakness of the network model strongly.
3. All Equations must typed with standred Equation Editor.
4.On page no.8, 3rd line, how the authors set the compression ratio of 16:1? Justify.
5.some Typo error see below
signif-icantly (Page No:4, line no.9)
Figure 1(Input not Iuput) 

6. The conclusion requires more focus and precise manner.

Comments on the Quality of English Language

Some contents were generated the LLM model. So, polish the English language with native English speaker

Author Response

(The authors gave the same response as above.)

Reviewer 3 Report

Comments and Suggestions for Authors

General concerns:

While the ms focuses on an important issue, there are major concerns regarding relevance of the experimental part of this study. While the authors start their Introduction focusing on the limitations of existing methods including advanced multispectral indices quoting them in a rather historical context, in their dataset description they refer to the masks arising just from those indices (more specifically, from NDWI), which sounds very inconsistent. The authors train ML models which they refer to the masks considered as ‘ground truth’, thus the best these models can do is adjust the outcome as close as possible to those masks. In turn, if the masks come from the same data which is being criticized in the first place, the overall design of the study becomes unclear and largely irrelevant. I would fully understand if the authors would, for example, use only monochrome or visible-band images, but refer to masks obtained with multispectral imaging for training and cross-validation of their approach. However, they also refer to the fusion of multispectral data in the context of ML approaches, which only adds to the confusion.

To summarize, the focus of the proposed approach should be more clearly stated and limited to specific scenarios where it is practically applicable. The key problem statement should be clearly positioned throughout the ms, stated explicitly already in the Abstract, continued in the Introduction, directly specified in the Materials and Methods which data are used for training, validation, and 'ground truth' masking, not just as a reference to an external previously published (sic!) dataset.

Specific suggestions:

- The introductory part on various ML designs including multiple versions with no direct relevance to this work appears largely redundant, as detailed discussions and comparisons of these approaches can be found in topical reviews and meta-analyses.

- Any claimed improvements should be supported by statistical significance evaluations with reference to tests and sample numbers.

Author Response

(The authors gave the same response as above.)

Reviewer 4 Report

Comments and Suggestions for Authors

This paper presents a novel framework for semantic segmentation of water bodies from satellite imagery. The authors propose a multi-faceted approach to improve accuracy and automate the model configuration process, which contains an enhanced U-Net architecture using a pre-trained ResNet50 as the encoder to improve feature extraction, a dynamic attention mechanism (an enhanced CBAM module) to adaptively calibrate spectral and spatial features, a hybrid evolutionary algorithm framework (integrating GA, ACO, PSO, and a proposed "Alpha Evolutionary" algorithm) to automate the optimization of key hyperparameters like learning rate, batch size, and momentum, and a learnable, weighted multi-loss function that combines Cross-Entropy/Focal loss with Dice loss to address class imbalance and improve boundary segmentation.
The authors validate their method on two public remote sensing datasets, demonstrating through comparative and ablation experiments that their proposed framework outperforms several baseline models across multiple evaluation metrics.
Here are my concerns.
1. The manuscript introduces four evolutionary algorithms (GA, ACO, PSO, and AE). While the first three are standard, "Alpha Evolutionary (AE)" is not defined, explained, or cited. Does this algorithm correspond to [1]? This is a critical omission. The authors must provide a detailed description of this algorithm or cite the relevant literature.
[1]. Hao Gao, Qingke Zhang, Alpha evolution: An efficient evolutionary algorithm with evolution path adaptation and matrix generation,
Engineering Applications of Artificial Intelligence, Volume 137, Part B, 2024, 109202, ISSN 0952-1976, https://doi.org/10.1016/j.engappai.2024.109202.
2. The mechanism for updating the weights of the different evolutionary algorithms over time (stages T1, T2, T3) is described with approximate values (e.g., "0.35±0.05"). This appears to be a pre-defined schedule rather than a truly dynamic or adaptive mechanism based on performance feedback. 
3. In Table 1, the results for the "VGG16+Adam" baseline are missing. Please provide the missing data or explain its absence.
4. The paper mentions two datasets (a Kaggle dataset and a Sentinel-2 dataset) but does not clearly delineate how they were used. Were they merged? Was one used for training and the other for testing to evaluate generalization? Details on data splitting (train/validation/test ratios) for each dataset are also missing.

Author Response

(The authors gave the same response as above.)

Round 2

Reviewer 1 Report

Comments and Suggestions for Authors

All the reviewer suggestions has been carefully incorporated and addressed in the revised manuscript.

Author Response

We sincerely thank for your thoughtful guidance and valuable time. Each of your suggestions has been deeply inspiring, and the transformation of this manuscript from the initial draft to the final version is entirely due to these valuable comments. We also hope to build on these insights to further plan future research directions and look forward to continued valuable achievements.

Reviewer 3 Report

Comments and Suggestions for Authors

The authors have addressed major concerns regarding the limitation of the study and described them explicitly in the Conclusion of the revised ms. Althogether, the revised ms could be potentially recommended for publication. While some parts of the Introduction have been edited, the Abstract remains to be overloaded by technical details referring to the well-known ML approaches that were employed here. Since they are only used as instruments, the suggestion is to shift the focus of the Abstract more to the problem and outcome of this current study and its potential applicability, rather than listing the (conventional) employed methods.

Author Response

Comments:

The authors have addressed major concerns regarding the limitation of the study and described them explicitly in the Conclusion of the revised ms. Alt hogether, the revised ms could be potentially recommended for publication. While some parts of the Introduction have been edited, the Abstract remains to be overloaded by technical details referring to the well-known ML approaches that were employed here. Since they are only used as instruments, the suggestion is to shift the focus of the Abstract more to the problem and outcome of this current study and its potential applicability, rather than listing the (conventional) employed methods.

Response

Thank you for your constructive suggestions. We have thoroughly revised the Abstract accordingly to shift the focus towards the research problem, key outcomes, and applicability. The technical details of the conventional methods have been streamlined, and greater emphasis is now placed on our fully automated framework, its performance gains, and its practical potential. The revised Abstract is provided below and in the manuscript. We believe it now better highlights the contribution of our work. If further adjustments are needed, we are happy to continue refining it based on your valuable feedback.

Revision (Page 1; Lines 16-30):

Accurate and automated identification of water bodies from satellite imagery is critical for environmental monitoring, water resource management, and disaster response. Current deep learning approaches, however, suffer from a strong dependence on manual hyperparameter tuning, which limits their automation capability and robustness in complex, multi-scale scenarios. To overcome this limitation, this study proposes a fully automated segmentation framework that synergistically integrates an enhanced U-Net model with a novel hybrid evolutionary optimization strategy. Extensive experiments on public Kaggle and Sentinel-2 datasets demonstrate the superior performance of our method, which achieves a Pixel Accuracy of 96.79% and an F1-Score of 94.75, outperforming various mainstream baseline models by over 10% in key metrics. The framework effectively ad-dresses the class imbalance problem and enhances feature representation without human intervention. This work provides a viable and efficient path toward fully automated re-mote sensing image analysis, with significant potential for application in large-scale water resource monitoring, dynamic environmental assessment, and emergency disaster management.

Reviewer 4 Report

Comments and Suggestions for Authors

I have no more comments. Congratulations

Author Response

We sincerely thank you for your thoughtful guidance and valuable time. Each of your suggestions has been deeply inspiring, and the transformation of this manuscript from the initial draft to the final version is entirely due to these valuable comments. We also hope to build on these insights to further plan future research directions and look forward to continued valuable achievements.